# Impact of the Soweto football derby on the trauma emergency department at Chris Hani Baragwanath Academic Hospital, a tertiary level hospital in South Africa

**Charles Baggott**[1]*, **Deirdré Kruger**[1], **Riaan Pretorius**[1,2]

**1** Faculty of Health Sciences, Department of Surgery, School of Clinical Medicine, University of the Witwatersrand, Johannesburg, South Africa, **2** Trauma Emergency Department, Chris Hani Baragwanath Academic Hospital, Soweto, South Africa

* dr.cbaggott@gmail.com

## Abstract

### Introduction

The Soweto Derby is one of Africa's largest football derbies. The two rival teams, Kaizer Chiefs and Orlando Pirates, both originate in Soweto, a sprawling township 20km outside Johannesburg. Soweto is infamous for the high levels of violent crime and trauma, but also for Chris Hani Baragwanath Academic Hospital (CHBAH), with one of the world's largest trauma emergency departments (ED). Research globally, describing the impact of sports events on public health care systems is conflicting, with evidence showing both increases and decreases in spectator related trauma. This paper seeks to describe the trauma burden during the Soweto Derby and add to the research concerning trauma relating to sporting derbies in low to middle income countries.

### Objectives

To analyze the impact of the Soweto Derby on the trauma ED at CHBAH over a 24-hour period.

### Methods

A retrospective comparative study at the CHBAH Trauma ED of 13 Soweto Derbies played over a 5 year period between 2015–2019, compared to the corresponding non-Soweto Derby days of the preceding year. Patients were triaged according to the South African Triage Scale and Advanced Trauma Life Support (ATLS) principles. Data was organized into 3 time frames where the triage score and mechanism of injuries were compared: 1) 4 hours pre-match, 2) 2 hours during the match, and 3) 18 hours post-match.

### Results

Thirteen Soweto Derbies and 2552 patients were included. The median age was 29 with males accounting for 73.4% of all trauma cases. Significantly more P1 patients presented

**Data Availability Statement:** All relevant data are within the manuscript and its Supporting information files. Data cannot be shared publicly because patient confidentiality has to be

maintained. Data are available from the University of the Witwatersrand Human Research Ethics Committee (contact www.witsethics.co.za/) for researchers who meet the criteria for access to confidential data.

**Funding:** The author(s) received no specific funding for this work.

**Competing interests:** The authors have declared that no competing interests exist.

during the Soweto Derby. Pre-match there were 3x less P1 patients presenting to the ED (4.7% vs 12%, p = 0.044). During the match, there was a 40% drop in males presenting to ED (5.95% vs 9.45%, p = 0.015). Post-match there was a significant increase in P1 patients treated (17.4% vs 13.5%, p = 0.021)), with the majority being young males. There was no increase in either female or paediatric visits to the ED.

## Conclusion

The Soweto Derby has a direct effect on the trauma burden at CHBAH, with more P1 patients presenting post-match. Young African males are disproportionally affected by severe trauma requiring increased health care resources in an already overburdened hospital.

## Introduction

South Africa is notorious for its high levels of violent crime and regularly cited as one of the most dangerous societies in the world. Exposure to trauma is commonplace and has negative implications on the quality of life for all South Africans, especially those living in poor socio-economic conditions [1,2]. Soweto, situated south of Johannesburg, was designed as a town-ship for Black people under the Apartheid system. Chris Hani Baragwanath Academic Hospi-tal (CHBAH) is the only tertiary level hospital that services Soweto's nearly 1.3 million residents. Some of the most violent areas of Soweto are located within a 10 km radius of CHBAH [3].

South Africa's largest football derby, known locally as the 'Soweto Derby', is one of the most recognized football derbies in Africa and draws avid fans from Soweto and neighbouring Johannesburg, as well as the rest of South Africa and abroad. The two football teams involved, Orlando Pirates and Kaizer Chiefs, are both Soweto based and combined attract the largest fan base in the country. CHBAH is situated less than 6.5 km from both home stadiums and is ide-ally placed to serve trauma patients impacted by the Derby either at the stadium or on televi-sion. No Soweto Derbies were played at the traditional home ground of Orlando Pirates, the Orlando Stadium during the research period, which is a much smaller stadium with a capacity of roughly 40000 spectators. By comparison, the FNB stadium accommodates 90000 spectators and is the stadium of choice for both teams due to the scope and magnitude of the event, with its increased capacity, modern facilities, and improved safety options.

Sporting events may have implications for Emergency Departments (EDs). Popular sport-ing events attract large numbers of people together, creating a sense of community. Con-versely, sport can generate heightened emotions, often fueled by increased alcohol consumption, which may result in irresponsible and aggressive behaviour. The pervading cul-ture of toxic masculinity coupled with the intergenerational legacy of oppression and political violence during Apartheid further contributes to Soweto's trauma burden [1,4]. CHBAH ED is already overwhelmed and may be under resourced to cope with increased trauma admissions related to the Soweto Derby.

Available international literature describing the relationship between trauma and prominent sporting events has been inconclusive, with conflicting evidence demonstrating both an increase and a decrease in sport-supporter related trauma [5–12]. Apart from one study which investigated the impact of trauma and mortality on the Cape Town pediatric population during

the 2010 FIFA World Cup, there has been no published literature showing the relationship between the trauma burden and large sporting events in low- to middle-income countries [12].

Therefore, the aim of this study was to determine the impact of the Soweto Derby on trauma patient admissions at CHBAH.

Specifically, we measured the overall burden of trauma during the Soweto Derby over three time periods: before, during and after the Soweto Derby. These time intervals were compared to trauma visits on corresponding non-derby weekends. Drawing on previous research and anecdotal opinions, we hypothesized that there would be an increase in trauma seen post-match.

## Methods

This retrospective comparative study was conducted at the Trauma ED of CHBAH.

The dates and times of the Soweto Derby from the years 2015–2019 were recorded. The first records from CHBAH ED were from the 2nd of August 2014 and the last date was on the 9th of November 2019.

The study population data was accessed between 01/08/2022 and 30/09/2022 from the Trauma ED register, recorded manually by triage nurses.

Demographics, triage score, vital signs, mechanism of injury and clinical diagnosis were collected for all trauma patients over a 24-hour period.

This was further divided into 3 intervals:

1. 4 hours *pre*-match

2. hours *during* the match, comprising two 45-minute halves and an additional 30 minutes consisting of a half-time break and extra-time

3. 18 hours *post*-match, commencing from the end of the match

Both adult and paediatric populations were included. Patients younger than 14 years of age were regarded as paediatric patients.

All medical and non-trauma related surgical patients were excluded.

The non-matchday control group was established by accessing data from the corresponding day of the week, during the same time, over the same month of the previous year. For example, the match played on the 1st Saturday of November 2019 (02/11/2019), was compared to the non-match day period during the first Saturday of November 2018 (03/11/2018). This 24-hour period factored in the same time frames; 4 hours pre-match, 2 hours during the match and 18 hours post-match.

The kickoff for all Soweto Derby matches is between 15h00 and 16h00 and concludes approximately 2 hours later. The objective of the control group was to accurately account for similar conditions including seasonal weather changes and other major events that may have occurred during the same period.

Patients were triaged according to ATLS principles and the South African Triage scale, which has been validated in both resource limited settings, as well as high income countries [13,14]. For the purposes of this study, patients were simplified as either P1 or non-P1. P1-patients required immediate emergency management in a resuscitation bay, while non-P1 patients, described traditionally as either P2 and P3, required urgent and non-urgent intervention respectively. Formal triage scores in the trauma setting often under-triage patients if too much emphasis is put on vital signs, as these are often late signs in an evolving pathology. Often the mechanism of injury is more important, and at CHBAH the mechanism of injury is weighted stronger than vital signs, irrespective of how stable the vital signs are.

Patients were occasionally triaged incorrectly. These were amended when reviewed by the doctor, who would upstage or downstage the patients accordingly. This is reflected in the nursing triage records.

The match day continuous data and the corresponding non-match day controls were compared using the Two-sample Wilcoxon rank-sum (Mann-Whitney), with medians and interquartile ranges (IQRs) reported. The Pearson's Chi-squared test and Fishers' exact test, where appropriate, were conducted for analyses of categorical data with absolute and relative frequencies reported. Statistical significance was considered for p-values below 0.05.

Permission to access all adult and paediatric data was approved by the CEO of CHBAH. Ethics approval was provided by the Human Research Ethics Committee of the Faculty of Health Sciences at the University of the Witwatersrand. No private information was gathered from the participants. All the participants details have been anonymized and therefore impossible for participants to be identified.

## Results

A total of 16 Soweto Derby matches were played during the study period of 2015–2019. Patient information was available for 13 of these matches, of which 11 matches were played at the FNB stadium. All these matches were broadcast live on television. A total of 26 days were assessed, comprising 13 Soweto Derby match days and 13 days of non-match day controls.

Demographic and patient characteristics are shown in Table 1 for the 2552 trauma ED patients included in the study according to whether their admission was during a Soweto Derby match day (n = 1432) or over a control weekend (n = 1120). In total, 312 more trauma patients were admitted on Soweto Derby match days, averaging 24 additional patients per match day, although this difference did not reach statistical significance.

The median (IQR) age was 29 (21.0–36.0) years and did not differ on match vs control day groups. The majority of patients were adults (85.1%) with a median (IQR) age of 30.0 (25.0–38.0) years, while the median (IQR) paediatric age was 6.0 (3.0–9.0) years, with no significant

Table 1. Demographic and clinical characteristics of the study population.

| Parameter | All patients | Match day group | Control day group | P value |
|---|---|---|---|---|
| | n = 2552 | n = 1432 | n = 1120 | |
| Age (years), median (IQR) | 29.0 (21.0–36.0) | 29 (22.0–36.0) | 28.5 (21.0–36.0) | 0.65 |
| Adults, n (%) | 2171 (85.1) | 1226 (85.6) | 945 (84.4) | 0.38 |
| Paediatrics, n (%) | 381 (14.9) | 206 (14.4) | 175 (15.6) | |
| Sex, n (%) | | | | |
| Male | 1895 (74.3) | 1059 (74.0) | 836 (74.6) | 0.69 |
| Female | 657 (25.7) | 373 (26.0) | 284 (25.4) | |
| Mechanism of injury, n (%) | | | | |
| Blunt | 1787 (70.0) | 1000 (69.8) | 787 (70.3) | 0.21 |
| Burns | 105 (4.1) | 51 (3.5) | 54 (4.8) | |
| Penetrating | 660 (25.9) | 381 (26.6) | 279 (24.9) | |
| Triage, n (%) | | | | |
| P1 | 343 (13.4) | 210 (14.7) | 133 (11.9) | 0.04 |
| Non-P1 | 2209 (86.6) | 1222 (85.3) | 987 (88.1) | |
| Intubated, n (%) | 46 (1.8) | 23 (1.6) | 23 (2.0) | 0.45 |
| Mortality, n (%) | 13 (0.51) | 6 (0.42) | 7 (0.62) | 0.58 |
| Road traffic accidents | 569 (22.3) | 323 (22.6) | 246 (22.0) | 0.72 |

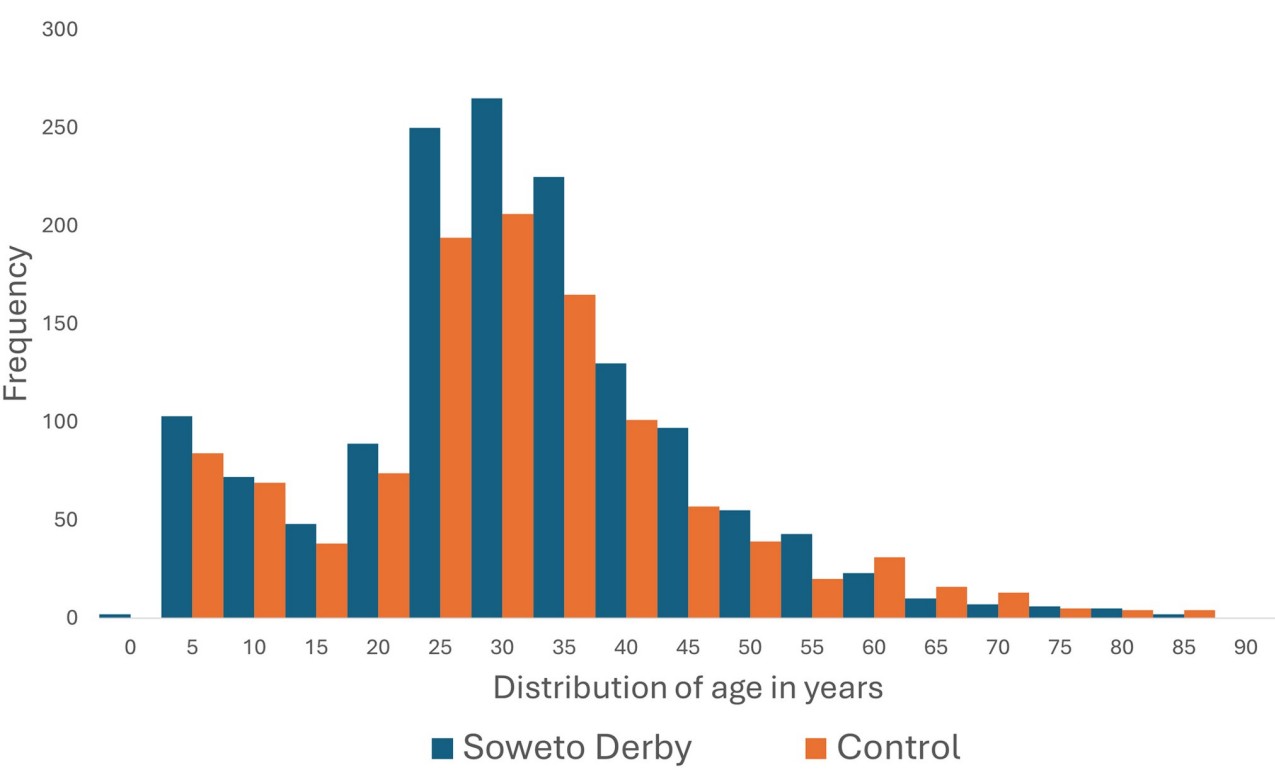

**Fig 1. Age distribution of all triaged patients during the Soweto Derby and those in the control group.**

differences found in match vs control day groups (see also Fig 1).74.3% of all patients were male, similar to the control and Soweto Derby groups (p = 0.69; Table 1).

Blunt injuries accounted for the majority of injuries (70.0%) followed by penetrating injuries (25.9%) and burns (4.1%) which did not deviate from the controls, demonstrating no statistical differences in mechanism of injury between Soweto Derby and controls.

From Table 1, there were more P1 patients treated on match days compared to control days (14.7% versus 11.9%, respectively; p = 0.04). However, this did not lead to an increase of intubated and ventilated patients in the resuscitation bays. Moreover, mortality rates in the ED were low (0.51%) and did not increase during Soweto Derby match days.

When analyzing the three time periods covering the 24 hours of Soweto Derby match days, there was no statistical difference in the overall number of patients treated during the time periods on Soweto Derby match days vs control days (p = 0.34). However, during the 4 hours pre-match, there was a statistical decrease in P1 patients treated on Soweto Derby days, with almost three times fewer P1 patients treated before the Soweto Derby (4.76% vs 12.03%, p = 0.044; Fig 2).

Similarly, during the match time period there was an almost 40% decrease in male patients presenting to CHBAH's Trauma ED during the Soweto Derby compared to control (5.95 vs 9.45%, p = 0.015; Fig 3), while there was no significant change recorded for either female or paediatric patients.

In the period 18 hours post-match, there was a significant increase in P1 patients on Soweto Derby days compared to the control group at 17.4% and 13.5%, respectively (p = 0.021).

There was no significant difference in the frequency of road accidents, either pedestrian vehicle collisions or motor-vehicle collisions between the two study groups, with 22.6% of

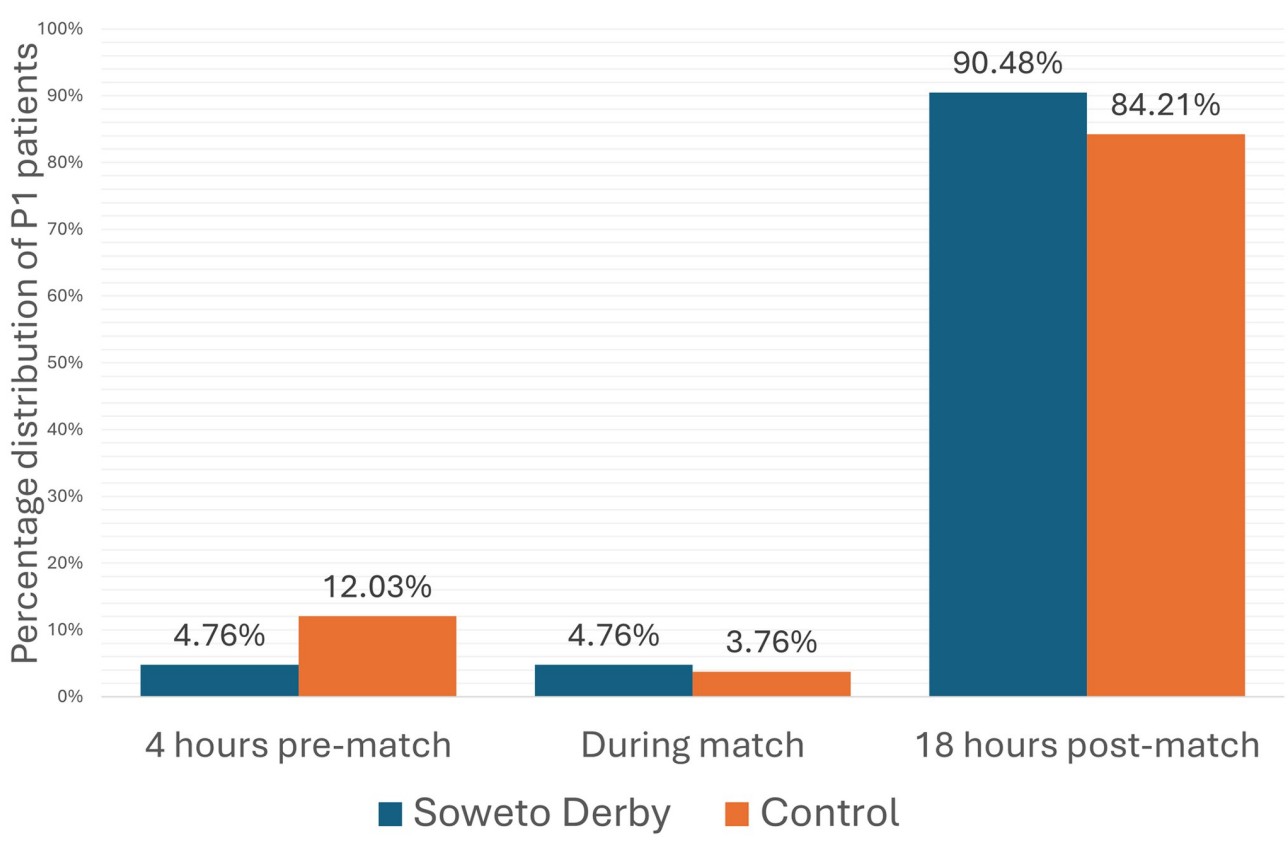

**Fig 2. Percentage of P1 admissions distributed over a 24-hour period during the Soweto Derby compared to the control group.**

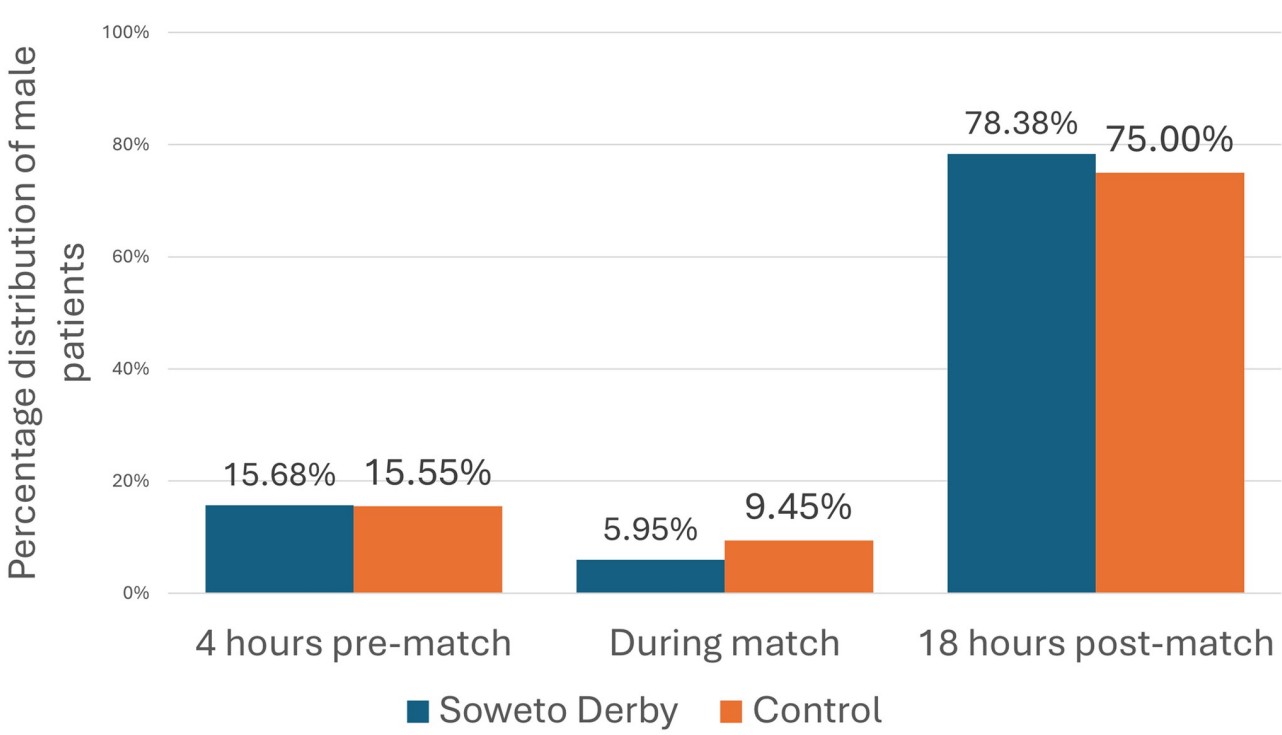

**Fig 3. Percentage of male patients distributed over a 24-hour period during the Soweto Derby days compared to the control group.**

trauma patients presenting with traffic-related accidents on match days compared to 22.0% in the control group (p = 0.72). However, an unexpected finding showed a significant decrease in traffic accidents when the result of the Soweto Derby ended in a draw, with 16.8% of trauma patients involved in a traffic-related accident, versus 23.4% and 25.4% when Kaizer Chiefs or Orlando Pirates won, respectively (p = 0.009).

## Discussion

There have been varied opinions amongst health care workers globally as to whether large-scale sporting events have a modulating effect on healthcare systems [5,7,8,15–17]. Our study is the first to assess the relationship between the trauma burden and large sporting events in a South African setting.

The literature indicates that the nature and frequency of ED visits are influenced by the timing of major football matches. Research internationally has shown a decrease in trauma-related casualties before and during matches, with a rebound increase of trauma-related incidents after a match [12,16,17]. This supports our findings.

By dividing the Soweto Derby day into three specific time periods, the nature and pattern of trauma visits to the ED at CHBAH was differentiated.

### Pre-match

There is limited research worldwide describing spectators' behaviour before major football matches. A Northern Irish study showed a significant decrease in adult ED attendances pre-match while a French paper and results from a Cape Town study showed significantly decreased paediatric visits to ED preceding major football matches [7,8,12].

Our results demonstrated no significant difference in paediatric visits preceding Soweto Derbies, supporting findings by Hughes et al which show a significant decrease in P1 patients treated at CHBAH before the Soweto Derby.

It is postulated that patient and parenteral behaviors change before major sporting events making them less inclined to seek urgent medical attention, fearing missing the match, which would result in fewer ED visits. Patients may also become distracted in the buildup to the match or avoid hospitals, concerned about hospital overcrowding [7,12].

### During the match

Our research showed a 40% decrease in men presenting to the Trauma ED during Soweto Derbies. International studies show that more than 65% of spectators are men. It is presumed in Soweto, more men watch the Soweto Derby, and hypothesized that male spectators shift their attention from their own ailments to the Soweto Derby, delaying hospital visits until after the match.

Intoxicated fans may only register the severity of their injuries after sobering up post-match [6,9]. In a retrospective study from Portugal examining Lisbon's largest football Derby, 20% less patients were treated during the match. Alesandrini et al reported that paediatric visits to French hospitals, including children accompanied by their fathers, decreased significantly when the French national team played [15].

### 18 hours post-match

There was a significant increase in P1 patients treated at CHBAH in the post-match period compared to the control group. A local study by Bhana et al showed Saturdays and Sundays accounted for 48% of trauma visits to CHBAH with the majority of patients treated between

18h00 and 06h00. This is consistent with this paper's findings where the18 hours post-match period were the busiest time for ED visits to CHBAH [3].

The Soweto Derby exacerbates problems seen every weekend which highlights the disproportionate levels of violence evident in this community, experienced mainly by young disadvantaged men, against the background of irresponsible alcohol use.

## Role of alcohol

Retrospective data on alcohol use was not available for this study, but there is a strong association worldwide between spectator binge drinking and sporting events [18,19]. This may be exacerbated in South African context, which has some of the highest per capita drinking rates globally. Two studies have shown that 45–59% of men treated in South African ED's are under the influence of alcohol [2,20,21]. Furthermore, disadvantaged communities like Soweto, suffer disproportionately from alcohol attributable morbidity and mortality rates [18,22]. The increased number of P1 patients treated after the Soweto Derby may be attributable to alcohol abuse and may account for the rebound effect of increased ED visits post-match [9,12,16,17]. Worldwide ED admissions during the COVID-19 pandemic dropped dramatically worldwide following national alcohol restrictions, emphasizing the dangerous role alcohol plays and the consequent impact on trauma visits [23].

## P1 admissions

International literature shows that the number of patients requiring hospitalization increases significantly during large sporting events, despite the total number of ED visits not being affected, which is also reflected in our findings [9]. The total number of patients seen in the CHBAH ED during the Soweto Derby was not statistically different to the control group, but there were significantly more severely affected P1 patients requiring increased hospital resources and interventions.

There are only 12 ventilators in the resuscitation bay at CHBAH ED which are all in use most Saturday nights. The increased P1 burden of Soweto Derbies may result in compromised patient care, as no further ventilators are available.

## Impact of gender and age on ED visits

The overwhelming majority of trauma patients in our research are men (74.3%). There was no significant change to paediatric or female ED visits during Soweto Derbies, contrary to international research, which shows an increase in domestic violence, primarily directed towards women and children during major sporting events [11,22].

## Match outcome and violence

Research has shown that match outcomes effects spectators' levels of aggression. Winning and upset losses result in more testosterone-related aggression and this leads to higher rates of drinking [18,22,24,25]. resulting in aggressive driving practices after winning and after upsets. Both Kaizer Chiefs and Orlando Pirates fans affected by trauma in Soweto would be seen at CHBAH, irrespective of the score line. A win for either team may lead to increased aggression and negligence on the road.

Our results point to a significant decrease in motor vehicle-related accidents when the match result was drawn, potentially due to a decrease in aggressive driving.

## Limitations

Patients were not followed up post-admission, consequently this paper is unable to establish whether there were increased mortality rates resulting from injuries relating to the Soweto Derby.

Ambulance services at CHBH are regularly overwhelmed on weekends [3]. Patients who were injured during the football match, may have called for an ambulance timeously, but constrained emergency resources may have resulted in patients receiving emergency services only after the match, thereby increasing patient load post-match.

Blood alcohol levels were not routinely tested. This information would have strengthened the research.

## Strengths

A total of 2552 trauma patient's records over six years were gathered. This large population strengthened the validity of the results.

The method of deriving the control group was an advantage.

Weekends account for 48% of all trauma presentations at CHBAH. Most ED visits at CHBAH occur on Saturdays between 18h00 and 06h00. The Soweto Derby is always played on a Saturday at the same starting time at 16h00 with the match ending around 18h00. This correlates to the busiest period of the week at CHBAH ED [3]. The control accounts for this.

The most violent areas in Soweto are located within a 10km radius of CHBAH [3]. As there aren't any other trauma facilities nearby, all serious trauma is directed to CHBAH. Therefore, the number of trauma cases seen at CHBAH reflects the actual trauma burden, which are exacerbated by the Soweto Derby.

## Conclusion

The Soweto Derby has a direct effect on the trauma burden at CHBAH. The rate and the type of trauma treated at CHBAH ED are statistically different before, during and after the Soweto Derby. There are less P1 patients treated before the derby, while during the derby, fewer males present to the ED. Conversely, there is a significant increase in the amount of P1 trauma at CHBAH post-match. Disadvantaged African males bear the brunt of violent trauma in Soweto. Neither women nor children's rates of trauma increased as a result of the Soweto Derby.

Further prospective studies are necessary to direct policy changes and improve collaboration between civil society and local government to more effectively address Soweto Derby spectator aggression. This information may also strengthen the planning and implementation of trauma responses at CHBAH, to make the Soweto Derby safer.

## Supporting information

**S1 Table. Types of injuries seen during the Soweto Derby and the control.**
(DOCX)

## Author Contributions

**Conceptualization:** Charles Baggott.

**Data curation:** Charles Baggott.

**Formal analysis:** Charles Baggott, Deirdré Kruger.

**Investigation:** Charles Baggott.

**Methodology:** Charles Baggott.

**Project administration:** Charles Baggott.

**Software:** Deirdré Kruger.

**Supervision:** Deirdré Kruger, Riaan Pretorius.

**Visualization:** Riaan Pretorius.

**Writing – original draft:** Charles Baggott.

**Writing – review & editing:** Charles Baggott, Deirdré Kruger.

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
