## [Decision Letter · Decision Letter 0]

12 Mar 2024

PONE-D-24-00705Impact of the Soweto Football Derby on the trauma emergency department at Chris Hani Baragwanath Academic Hospital, a tertiary level hospital in South AfricaPLOS ONE

Dear Dr. Baggott,

Thank you for submitting your manuscript to PLOS ONE. After careful consideration, we feel that it has merit but does not fully meet PLOS ONE’s publication criteria as it currently stands. Therefore, we invite you to submit a revised version of the manuscript that addresses the points raised during the review process.

We look forward to receiving your revised manuscript.

Kind regards,

Donovan Anthony McGrowder, PhD., MA., MSc

Academic Editor

PLOS ONE

Journal Requirements:

2. We note that your Data Availability Statement is currently as follows: All relevant data are within the manuscript and its Supporting Information files

4. Please remove your figures from within your manuscript file, leaving only the individual TIFF/EPS image files, uploaded separately. These will be automatically included in the reviewers’ PDF.

Additional Editor Comments:

Dear Dr. Baddott,

 Your manuscript “Impact of the Soweto Football Derby on the trauma emergency department at Chris Hani Baragwanath Academic Hospital, a tertiary level hospital in South Africa” has been assessed by our reviewers. They have raised a number of points which we believe would improve the manuscript and may allow a revised version to be published in PLOS ONE. Their reports, together with any other comments given are below.

 If you are able to fully address these points, we would encourage you to submit a revised manuscript to PLOS ONE by the date given below.

 Best regards,

Dr. Donovan McGrowder

Reviewers' comments:

Reviewer's Responses to Questions

**Comments to the Author**

1. Is the manuscript technically sound, and do the data support the conclusions?

Reviewer #1: Yes

Reviewer #2: Yes

2. Has the statistical analysis been performed appropriately and rigorously? 

Reviewer #1: Yes

Reviewer #2: Yes

3. Have the authors made all data underlying the findings in their manuscript fully available?

Reviewer #1: Yes

Reviewer #2: Yes

4. Is the manuscript presented in an intelligible fashion and written in standard English?

Reviewer #1: Yes

Reviewer #2: Yes

5. Review Comments to the Author

**Reviewer #1:** Dear colleagues,

thank you very much for allowing me to review your important wor. You have worked on an very interesting topic both LMIC and high income countries, but maybe has a higher burden on heath care systems with restricted resources.

In your results, please elaborate more in detail on the respective injuries and on outcome of the patients in both groups. Currently, no details on injuries and patients condition, as well as interventions were reported. Secondly, please consider to perform a matched pairs analyses between your intervention and your standard group with e.g. mortality in the ED as your common endpoint. By doing so, you would give the reader a better and broader picture of what happend in your department.

Furthermore I have some more minor comments. First, please re-write the methods section in your abstract since currently it is not easy to understand. Secondly, figure 2 is not easy to understand as well. Please make sure that all figures are exactly showing what you want to demonstrate to the reader. Thirdly, in the results section you have described one of the two stadiums. Please give more details tio the second one.

I am very much looking forward to see the revised version of the manucript.

Kind regards

**Reviewer #2:** Dear authors,

I have read your manuscript with great interest. I think it nicely adds to existing literature. However, there are a few points I recommend to consider before publication:

-) Abstract, general: Many of your sentences are either too vague or contain information which must be confusing to someone reading this for the first time. For example: "Determining the impact" of something hardly describes your methodology. "P1" patients is not defined. Why do you speak of a derby as one event on the one hand and then several matches on the other?

-) Abstract, specific: "No research exists in low-to-middle income countries and this research hopes to bridge this gap." - Please rephrase. I'm sure you don't mean there is no research at all in such countries.

-) Comparing patients by their triage category can be tricky. From own experience, your nursing staff doing the triage is not always doing this accurately or thoroughly, leaving a high risk of bias here. Maybe try re-categorizing your patients by using their vital signs etc.? Or at least do a sample cross check if triaing was correct?

-) Figures 1-3: There is far too little information in the figure legends. Please explain all abbreviations and give some context.

-) The discussion section is rather long; maybe try steamlining it and introduce subheadings to group trains of thought.

-) The conclusion should be shorter and more concise.

6. PLOS authors have the option to publish the peer review history of their article (what does this mean?). If published, this will include your full peer review and any attached files.

Reviewer #1: No

Reviewer #2: No

---

## [Author Response · Author response to Decision Letter 0]

23 Apr 2024

Rebuttal Letter

Manuscript reference number: PONE-D-24-00705

Title: Impact of the Soweto Football Derby on the trauma emergency department at Chris Hani Baragwanath Academic Hospital, a tertiary level hospital in South Africa

Dear Reviewers, 

Thank you for the valuable and constructive comments and suggestions which we believe have led to an improved revised manuscript. 

We hope that this Rebuttal Letter will be able to address all the questions that were raised.

All changes are in relation to the document with track changes, set on “All Markup” with “Show Revisions in Balloons”.

In response to the comments on the general journal requirements:

Response: We have gone through the PLOS ONE’s style requirements and have edited the body of work which needed to be in line with the journal’s required style. 

2) We note that your Data Availability Statement is currently as follows: All relevant data are within the manuscript and its Supporting Information files. Please confirm at this time whether or not your submission contains all raw data required to replicate the results of your study. Authors must share the “minimal data set” for their submission. PLOS defines the minimal data set to consist of the data required to replicate all study findings reported in the article, as well as related metadata and methods.

Response: All our data is available. However, our national ethics approval certificates do not allow us to share any raw data without a legal data transfer agreement in place. Data cannot be shared publicly because patient confidentiality must be maintained. Data are available from the University of the Witwatersrand Human Research Ethics committee (contact www.witsethics.co.za/) for researchers who meet the criteria for access to confidential data. 

3) We note that you have included the phrase “data not shown” in your manuscript. Unfortunately, this does not meet our data sharing requirements. PLOS does not permit references to inaccessible data. We require that authors provide all relevant data within the paper, Supporting Information files, or in an acceptable, public repository. Please add a citation to support this phrase or upload the data that corresponds with these findings to a stable repository (such as Figshare or Dryad) and provide and URLs, DOIs, or accession numbers that may be used to access these data. Or, if the data are not a core part of the research being presented in your study, we ask that you remove the phrase that refers to these data.

Response: Thank you for bringing this to our attention. In the process of cleaning up our results and discussion we felt that knowing that there were differences in sex according to age did not add to our research and has been deemed not a core part of the research, especially because most patients seen were young males. The relevant information regarding gender and age has already been added and discussed in the text.

Response to Reviewer #1

Comments: 

1) In your results, please elaborate more in detail on the respective injuries and on outcome of the patients in both groups. Currently, no details on injuries and patients condition, as well as interventions were reported. 

Response: In our work, we have categorized the trauma injuries as either blunt, penetrating or burns, which is a way of differentiating the patients succinctly in the trauma environment. In retrospect, detailing the specific types of injuries is very interesting. We have added a supplementary table (S1 Table 1) which has broken down the injuries into the main types, while at the same time keeping the injuries categorized into the three main categories (namely blunt, penetrating and burns). Please see the Supporting information S1 Table 1 on page 17 line 1533.

Additionally, patients were categorized according to their triage score, which gives an indication of the patient’s condition. This is reflected in Table 1 on page 9 line 282.

One of the limitations to this retrospective study is that patients were not followed up after the 24-hour period, and so we could not account for patients interventions outside of the ED, nor could we account for the patients who subsequently died while either admitted in the ward or in theatre. See limitations on page 14 line 1085-1094.

2) Please consider to perform a matched pairs analyses between your intervention and your standard group with e.g. mortality in the ED as your common endpoint. By doing so, you would give the reader a better and broader picture of what happened in your department. Response: Thank you for this comment. As part of our preliminary data analysis we had done an age dependent matched paired analysis between the Soweto Derby and the control. This did not change any of our significant findings, especially with no change in P1 patients seen overall or when divided into before the match, during the match, or after the match. Mortality was also assessed as an end point in the ED was and has been commented on page 8 line 259-262 and is represented in Table 1 page 9 line 282. 

However, there was no significant difference in mortality and intubated patients between the Soweto Derby and the control, even with match paired analysis. Additionally, the data showed that the number of trauma patients intubated in the ED also did not increase. This was a surprising result for us as we had assumed that with a significant increase in P1 patients we would expect to see more intubations. However, on busy Saturday nights, in both the Soweto Derby and the controls, the 12 ventilators used in casualty were mostly in use, and there wasn’t any capacity to use any more, and this could possibly be the reason why there wasn’t a significant increase in intubations, even though it could have potentially led to impaired patient care. This has been elaborated on page 13 line 766-768.

3) Please re-write the methods section in your abstract since currently it is not easy to understand. 

Response: The methods section in the abstract has been rewritten as requested. We hope this removes all ambiguity in both the abstract as well as in the methodology section. See page 2 line 32-37.

4) Figure 2 is not easy to understand as well. Please make sure that all figures are exactly showing what you want to demonstrate to the reader.

Response: Figure 2 has been replaced with a Bar Graph which we feel is easier to understand. The legend as well as the X and Y axis have been updated and better describe our findings. The figures have been removed from the manuscript as per PLOS 1 guidelines. Please see the figures in the attached TIFF image.

5) The results section you have described one of the two stadiums. Please give more details of the second one.

Response: The home stadium of Orlando Pirates is called Orlando Stadium. It is also situated in Soweto, also less than 10km from Chris Hani Baragwanath Academic Hospital. 

The home stadium for Kaizer Chiefs is FNB stadium, which is also less than 10km from Chris Hani Baragwanath Academic Hospital. In the research period, Orlando Stadium wasn’t used during the Soweto Derby. This is because it is a much smaller stadium, which accommodates roughly 40000 spectators. FNB stadium has a capacity of more than 90000 and is one of the largest football stadiums in the world. Both teams treat this fixture as a home game as the spectators are coming from the same areas of Soweto and surrounds. Economically for both clubs mean that Orlando Stadium during the Soweto Derby is obsolete. This has been reflected in the revised results section. This has been discussed on page 4 line 125-129.

Reviewer #2 

1) Abstract, general: Many of your sentences are either too vague or contain information which must be confusing to someone reading this for the first time. For example: "Determining the impact" of something hardly describes your methodology. 

Response: The abstract has been rewritten, taking into account the comments of reviewer #1 and #2. See page 2 and 3

2) "P1" patients is not defined.

Response: P1 patients are patients that require immediate emergency care and are relatively uniform in most trauma triage systems. Chris Hani Baragwanath Academic Hospital has one of the busiest Trauma ED in the world, and makes use of triage scores based on ATLS principles and a modified version of the South African Triage Scale to help triage P1 patients correctly. This particular triage system has been validated in both LMIC and 1st world countries. This has been amended in the revised document. See page 2 lines32-37 and page 7 lines 210-221. 

3) Why do you speak of a derby as one event on the one hand and then several matches on the other. 

Response: Thank you brining this to our attention. All matches described in the text relate to the Soweto Derby. The Soweto Derby is a specific football match played between Kaizer Chiefs and Orlando Pirates, and in the context of that, all matches discussed in this paper refer to this particular football derby. We have removed ‘matches’, and replaced it with ‘Soweto Derby’ throughout to minimize any ambiguity. 

4) Abstract, specific: "No research exists in low-to-middle income countries and this research hopes to bridge this gap." - Please rephrase. I'm sure you don't mean there is no research at all in such countries. 

Response: The abstract has been rewritten to better describe our research. You are correct, this particular research is not the only research in LMIC, however it is the first one that we have seen that directly compares a domestic football derby and trauma in a LMIC. The only other research to our knowledge that was similar to this were done in Spain and Portugal, both of which are not LMIC. See abstract on page 2-3 lines 21-103.

5) Comparing patients by their triage category can be tricky. From own experience, your nursing staff doing the triage is not always doing this accurately or thoroughly, leaving a high risk of bias here. Maybe try re-categorizing your patients by using their vital signs etc.? Or at least do a sample cross check if triaing was correct? 

Response: Comparing patients by their triage score helps compare like vs like. Patients are triaged by both nurses and doctors, and when there is a discrepancy then this is reflected in the triage files when patients are stepped up to the resuscitation bays or stepped down to the general trauma pit. All patients have to be accounted for in the triage file, whether admitted to hospital, discharged or sent to the mortuary. This cross checking of all patients decreases the chance of patients falling through the cracks. There is a very large sample size of 2552 patients in this study, and this would also decrease the potential bias of poorly triaged patients. Using vital signs alone cannot accurately account for all P1 patients in the trauma setting, as often vital sign changes are late signs. Often more importantly, patients are declared P1 by the mechanism of injury. For example, all gunshot wounds to the abdomen or chest are P1 patients irrespective of how stable the vital signs are. This has been further elaborated on and reflects in the methodology on page 5 lines 162-231.

6) Figures 1-3: There is far too little information in the figure legends. Please explain all abbreviations and give some context. 

Response: All figures have been redone to better depict the relevant statistics. All figures are removed and added separately as per PLOS 1 protocol. Please see the attached TIFF image files.

7) The discussion section is rather long; maybe try steamlining it and introduce subheadings to group trains of thought. 

Response: Thank you for bringing this to our attention. Upon reflection it is far too long. The discussion has been edited and is more concise. Subheadings have also been used to help with the general flow. Please see the discussion on pages 10-14 lines 323-1084. 

8) The conclusion should be shorter and more concise. 

Response: The conclusion has also been edited, and has become shorter and more concise. Please see page 15 lines 1299-1309.

Thank you for the valuable feedback in improving our manuscript and the opportunity to resubmit.

Yours sincerely,

Dr Baggott, Prof Kruger and Dr Pretorius

---

## [Decision Letter · Decision Letter 1]

25 Jun 2024

Impact of the Soweto Football Derby on the trauma emergency department at Chris Hani Baragwanath Academic Hospital, a tertiary level hospital in South Africa

PONE-D-24-00705R1

Dear Dr. Baggott,

We’re pleased to inform you that your manuscript has been judged scientifically suitable for publication and will be formally accepted for publication once it meets all outstanding technical requirements.

Kind regards,

Uday Bhaskar Manda

Academic Editor

PLOS ONE

Additional Editor Comments (optional):

Dear Dr. Baggott,

Upon reviewing the comments from the reviewers I am pleased to inform that your manuscript has been accepted.

Thanks,

Uday Manda

Academic Editor.

Reviewers' comments:

Reviewer's Responses to Questions

**Comments to the Author**

1. If the authors have adequately addressed your comments raised in a previous round of review and you feel that this manuscript is now acceptable for publication, you may indicate that here to bypass the “Comments to the Author” section, enter your conflict of interest statement in the “Confidential to Editor” section, and submit your "Accept" recommendation.

Reviewer #1: All comments have been addressed

Reviewer #2: All comments have been addressed

2. Is the manuscript technically sound, and do the data support the conclusions?

Reviewer #1: Yes

Reviewer #2: Yes

3. Has the statistical analysis been performed appropriately and rigorously? 

Reviewer #1: Yes

Reviewer #2: Yes

4. Have the authors made all data underlying the findings in their manuscript fully available?

Reviewer #1: Yes

Reviewer #2: Yes

5. Is the manuscript presented in an intelligible fashion and written in standard English?

Reviewer #1: Yes

Reviewer #2: Yes

6. Review Comments to the Author

Reviewer #1: Dear colleagues,

thanks a lot for adressing all my questions and concerns. I have no further questions.

Kind regards,

Reviewer #2: (No Response)

7. PLOS authors have the option to publish the peer review history of their article (what does this mean?). If published, this will include your full peer review and any attached files.

Reviewer #1: No

Reviewer #2: No

---

## [Editor Report · Acceptance letter]

2 Jul 2024

PONE-D-24-00705R1 

PLOS ONE

Dear Dr. Baggott, 

I'm pleased to inform you that your manuscript has been deemed suitable for publication in PLOS ONE. Congratulations! Your manuscript is now being handed over to our production team.

Kind regards, 

on behalf of

Dr. Uday Bhaskar Manda 

Academic Editor

PLOS ONE